# Identifying Soil Erosion Processes in Alpine Grasslands on Aerial Imagery with a U-Net Convolutional Neural Network

**Maxim Samarin [1,*,†], Lauren Zweifel [2,†], Volker Roth [1] and Christine Alewell [2]**

[1] Department of Mathematics and Computer Science, University of Basel, Spiegelgasse 1, 4051 Basel, Switzerland; volker.roth@unibas.ch

[2] Department of Environmental Sciences, University of Basel, Bernoullistrasse 30, 4056 Basel, Switzerland; lauren.zweifel@unibas.ch (L.Z.); christine.alewell@unibas.ch (C.A.)

* Correspondence: maxim.samarin@unibas.ch
† These authors contributed equally to this work.

**Abstract:** Erosion in alpine grasslands is a major threat to ecosystem services of alpine soils. Natural causes for the occurrence of soil erosion are steep topography and prevailing climate conditions in combination with soil fragility. To increase our understanding of ongoing erosion processes and support sustainable land-use management, there is a need to acquire detailed information on spatial occurrence and temporal trends. Existing approaches to identify these trends are typically laborious, have lack of transferability to other regions, and are consequently only applicable to smaller regions. In order to overcome these limitations and create a sophisticated erosion monitoring tool capable of large-scale analysis, we developed a model based on U-Net, a fully convolutional neural network, to map different erosion processes on high-resolution aerial images (RGB, 0.25–0.5 m). U-Net was trained on a high-quality data set consisting of labeled erosion sites mapped with object-based image analysis (OBIA) for the Urseren Valley (Central Swiss Alps) for five aerial images (16 year period). We used the U-Net model to map the same study area and conduct quality assessments based on a held-out test region and a temporal transferability test on new images. Erosion classes are assigned according to their type (shallow landslide and sites with reduced vegetation affected by sheet erosion) or land-use impacts (livestock trails and larger management affected areas). We show that results obtained by OBIA and U-Net follow similar linear trends for the 16 year study period, exhibiting increases in total degraded area of 167% and 201%, respectively. Segmentations of eroded sites are generally in good agreement, but also display method-specific differences, which lead to an overall precision of 73%, a recall of 84%, and a $F_1$-score of 78%. Our results show that U-Net is transferable to spatially (within our study area) and temporally unseen data (data from new years) and is therefore a method suitable to efficiently and successfully capture the temporal trends and spatial heterogeneity of degradation in alpine grasslands. Additionally, U-Net is a powerful and robust tool to map erosion sites in a predictive manner utilising large amounts of new aerial imagery.

**Keywords:** deep learning; semantic segmentation; remote sensing; object-based image analysis; erosion mapping; landslides; livestock trails; sheet erosion

## 1. Introduction

Soil degradation is a major ecological threat which affects many areas of the world and can be accelerated by land-use management and changing climate parameters, such as precipitation and temperature [1–5]. In Switzerland, some alpine grassland areas are strongly affected by soil erosion

due to the steep terrain and extreme climate conditions. While soil erosion occurs naturally in these environments—in the form of landslides (triggered by snow gliding or heavy precipitation events) or sheet erosion (the process of the removal of topsoil caused by rain drops' impacts and overland flow)—there are also anthropogenic influences (e.g., agricultural activities) which can accelerate erosion rates [3,6,7]. For example, livestock keeping can lead to overgrazing and trampling in favoured grazing areas. Over time, livestock trails develop and trampling and grazing can lead to a reduction in vegetation cover, which in turn is prone to sheet erosion [8,9]. Additionally, livestock keeping can cause instabilities on slopes and ultimately result in landslides [10].

Therefore, erosion processes have strong temporal and spatial dynamic components, which is why large-scale understanding and detailed mapping over time and space is of great importance for long-term sustainable management practices.

Alpine areas are difficult to access and erosion features can affect substantial areas, making a comprehensive understanding of ongoing erosion processes unattainable from the ground. Larger-scale erosion studies for Switzerland have mainly been approached with the help of soil erosion modelling—e.g., the (revised) universal soil loss equation [11–18]. To achieve a thorough understanding of potential soil erosion threats, it is important to combine model outputs with observations for validation purposes [19]. The latter is especially crucial in mountainous and grassland areas, where model suitability has been questioned (see discussion in Alewell et al. [20]). High-resolution aerial imagery offers the opportunity to remotely assess and map the spatial extent of bare soil sites and sites with strongly reduced vegetation cover, allowing certain constraints to be overcome, such as the inaccessibility or extent of a study area. Object-based image analysis (OBIA) is an approach commonly used to identify urban and natural "objects" on satellite and aerial imagery and has been successfully used in the past to map various forms of soil erosion [7,21–30]. OBIA creates image segments by grouping pixels with similar properties together, which can then be classified based on object information (spectral, spatial, textural, and contextual) with expertly developed classification rules and/or various machine learning classifiers. OBIA is a method suitable for smaller study areas, but large-scale studies become difficult to manage. Limitations including processing times, a lack of work-flow transferability to other scenes, and the involvement of manual steps hinder efficient spatial up-scaling of projects. In past years, deep learning methods have progressively been applied in the field of remote sensing for image classification tasks and segmentation tasks [31–34]. In this study, we apply a deep learning method to demonstrate that it is capable of mapping and classifying soil erosion features on aerial images in a fast, objective, reliable, and scalable manner. We apply a fully-convolutional neural network (CNN) framework using the U-Net architecture developed by Ronneberger et al. [35]. In general, the U-Net architecture offers itself to semantic segmentation tasks with limited training data. U-Net and variations of this architecture have become increasingly popular for remote sensing tasks. Many applications focus on urban settings for road [36–39] or building extraction [40–43] from satellite and aerial imagery. Applications in a natural environment are constrained by the limited availability of high-quality labelled training data. Despite this limitation, U-Net has been applied in cloud detection on satellite images [44], mapping of woody vegetation [45], segmentation of plant species [46], forest damage assessment [47], the extraction of Antarctic glacier and ice shelf fronts [48], and archaeological studies [49] to name a few. Our annotated training data has been generated by mapping erosion sites on aerial images using OBIA for a valley in the Central Swiss Alps (Urseren Valley, Canton of Uri). We compare U-Net results to OBIA mapping for a held-out test region (area of 17 km$^2$), which was not used for training (9 km$^2$) for the years 2000, 2004, 2010, and 2013. Additionally, we investigate both the temporal and the spatial transferability of the U-Net method by mapping a new aerial image not seen during training (2016). Our main objectives of this study are: firstly, to show that the fully automated U-Net approach is capable of reproducing the high-quality soil erosion mapping and the temporal trends as they were attained with OBIA for the same study site; secondly, to show that the U-Net approach generalises well to new aerial images, i.e., can be used in a predictive manner to perform adequate segmentation of previously unseen input

data. In contrast, the OBIA procedure typically eludes such predictive usage and needs to be adjusted for each new aerial image. The capabilities and the fully automated nature of the U-Net approach make it a highly promising tool for efficient large-scale erosion mapping (e.g., alpine-wide analysis of soil erosion in semi-natural ecosystems such as grasslands and bush-land).

## 2. Study Area

The Urseren Valley (26 km$^2$) is an alpine valley located in Central Switzerland in the southern part of the Canton of Uri (Figure 1). The valley has a NE–SW orientation, and exhibits steep slopes (average angle of 27°) and rough terrain. The valley is geologically divided into two distinct sections and separated by the river Reuss: The northern slope is part of the Aarmassif (granite), and the southern slope belongs to the Gotthard massif (gneiss). Located between these two massifs near the valley floor is the so-called Urseren—Garvera zone (Mesozoic sediments) [50]. The dominant soil types in the catchment are Podzols and Cambisols, with Leptosols commonly found on steep slopes (classified after IUSS Working Group WRB [51]).

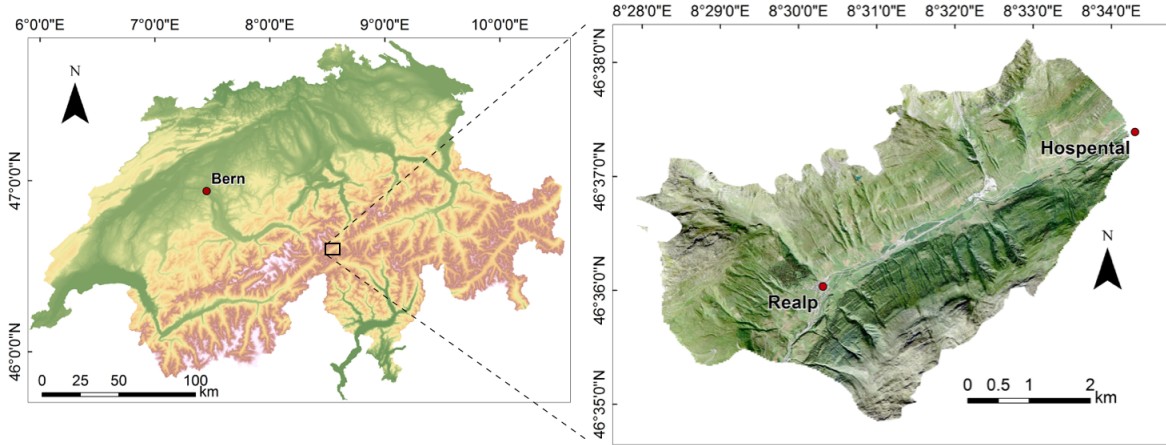

**Figure 1.** The Urseren Valley is located in the Central Swiss Alps in the Canton of Uri. The left map contains the topographic map of Switzerland (from low elevations in green to high elevations in brown to white). The right image contains an aerial image of the Urseren Valley overlaid on a hill-shade map of the area.

The 30 year average temperature (1990–2019) of the closest meteorological station in Andermatt (1438 m a.s.l.) is 3.9 °C. The average temperature has increased by 0.7 °C during the last 10 years (compared to the average of 1980–2009). The average rainfall during the last 30 years was 1384 mm with an average maximum 3 day precipitation intensity of 123 mm/3 d. The average seasonal (November–April) snow height is 58 cm with maximum snow heights during February/March (average of 103 cm) (data provided by MeteoSwiss, 2020). The dominant land-covers are grassland (including dwarf-shrubs consisting of Calluna vulgaris, Rhododendron ferrugineum, and Juniperus sibirica), which is mainly used for grazing (i.e., sheep and cattle) and haying, shrubs (mainly Alnus viridis and Sorbus aucuparia), and debris/bare rock areas [3]. Shrub encroachment due to land abandonment and extensification is present in the valley. Avalanches and snow gliding occur frequently in the Urseren Valley, facilitated by the deforested state of the slopes. The dominant erosion processes in this region are (shallow) landslides, sheet erosion, and erosion caused by land-use management (livestock, machinery, and manuring). Additional information on the Urseren Valley and occurring erosion processes can be found in Meusburger and Alewell [3], Zweifel et al. [7], Alewell et al. [52].

## 3. Data Sets

In the following we present the data sets used in our study. Table 1 summarises the data sets used for the mapping procedure conducted with U-Net which were also the basis for the training data set produced with OBIA [7].

**Table 1.** Summary of raster data sets used in this study. All geodata sets© Swisstopo. *: The aerial image of 2016 was only used for validation purposes of the U-Net model.

| Data Set | Derivative | Spectral Bands | Spatial Res. | Recording Date | |
|---|---|---|---|---|---|
| Aerial Image | | Red, Green, Blue | 0.5 m | 24 August | 2000 |
| | | Red, Green, Blue | 0.5 m | 9 September | 2004 |
| | | Red, Green, Blue | 0.25 m | 20 July | 2010 |
| | | Red, Green, Blue | 0.25 m | 1 August | 2013 |
| | | Red, Green, Blue | 0.25 m | 20 July | 2016 * |
| Digital Terrain | Slope | | 2 m | | |
| Model (DTM) | Aspect | | 2 m | | |
| | Curvature | | 2 m | | |

### 3.1. Aerial Imagery

The aerial images of SwissImage are high-resolution georeferenced orthophotos (product of Swisstopo [53]). Five aerial images covering the Urseren Valley were used in the time from 2000 to 2016. These images have a spatial resolution of 0.5 or 0.25 m (Table 1). Spectral information is available in the visible range (red, green, and blue spectral bands). All aerial images have slightly different properties (e.g., spatial resolution, colour distribution, and lighting conditions) but were always recorded during the growing season between late July and early September.

### 3.2. Digital Terrain Model

The digital terrain model (DTM) SwissALTI3D is the surface model of Switzerland without vegetation and development and has a spatial resolution of 2 m (product of Swisstopo [54]). Based on the elevation information of the DTM we derived the slope, aspect, and curvature (plan and profile) using ArcGIS (Version 10.5). The DTM provides valuable information and offers context to the aerial images. Zweifel et al. [7] have shown that for their study using OBIA, the DTM and its derivatives were essential for successful erosion mapping and classification.

### 3.3. Training Data

The data used to train the U-Net model consists of aerial imagery, DTM information, and training labels (see Section 4.3 for the training process). To train our U-Net model, a subsection (9 km$^2$) of the Urseren Valley (26 km$^2$) was used with the corresponding OBIA-mapped features (Figure 2). Four of the aerial images were used during training, leaving out the year 2016. By separating a subsection for training, we tested the spatial transferability of the model within the larger valley region. In addition, by omitting 2016, we investigated the spatial and temporal transferability when applying U-Net to a different image with properties not known during training.

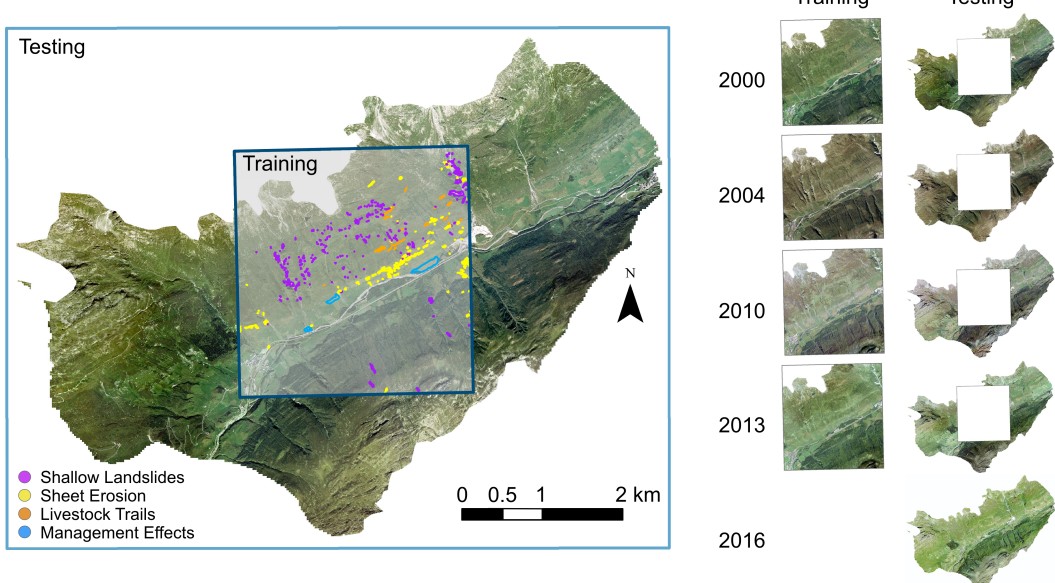

**Figure 2.** Training (9 km²) and testing (17 km²) areas are marked on the aerial image with examples of OBIA training labels for 2000 (map on the left). On the right-hand side is an overview of all available years and the sections used for training and testing. All training areas contain OBIA training labels (not shown) for the respective years (2000–2013). Training labels vary for each year due to the continuous evolution of soil erosion sites. The entire area of the image taken in 2016 was used only for testing.

### 3.3.1. Training Labels

The training labels come in the form of mapped erosion sites with attributed erosion classes from a previous study by Zweifel et al. [7]. This data set was created with a semi-automatic method using an OBIA approach described in Section 4.1, which made use of the same aerial imagery and DTM information as used for U-Net. Mapped erosion objects are available for the entire Urseren Valley for all five aerial images (2000, 2004, 2010, 2013, and 2016). Based on random sample evaluation by experts, this data set has an average overall accuracy score of 85.4% [7]. The training labels consist of four different erosion classes: shallow landslides (areas with displaced topsoil layers and clear boundaries to the surrounding vegetation), livestock trails (elongated tracks caused by livestock trampling, mostly perpendicular to the slope), sheet erosion (patches with reduced vegetation cover), and management effects (large areas damaged by heavy machinery, over-fertilisation, or intense grazing in fenced-off areas) (see Figure 3).

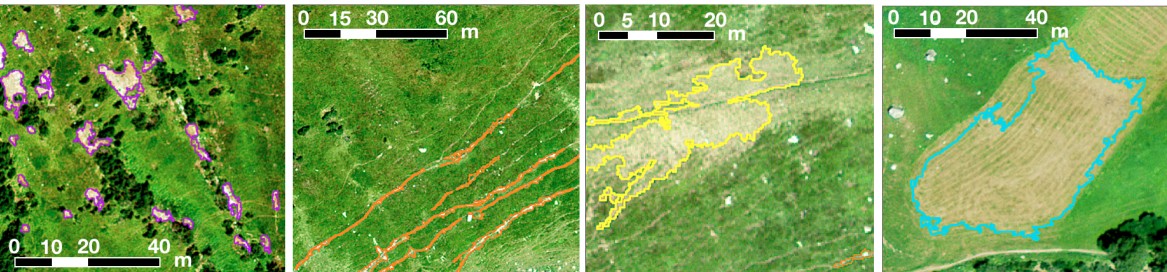

**Figure 3.** Examples for the labels used for training the U-Net model. From left to right: shallow landslides, livestock trails, sheet erosion, management effects.

## 4. Methodology

Our methodology consists of two major parts: the training process and the prediction process, with an overview depicted in Figure 4. To train the U-Net model we use OBIA labels together with the respective aerial image information (RGB) and DTM information for a dedicated training area (9 km$^2$). U-Net assigns pixel-wise probability values and thus provides information about the likelihood of pixels belonging to a specific erosion class. Based on these probabilistic assignments, hard segmentations are produced by thresholding. The following sections will describe the methodology in further detail.

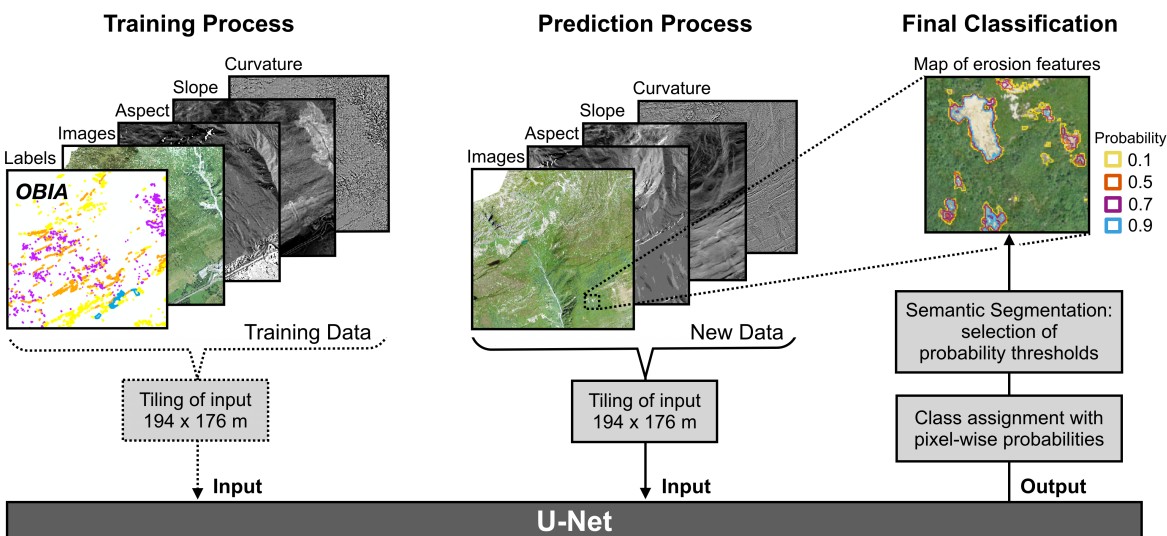

**Figure 4.** An overview of the developed workflow on the basis of U-Net showing examples of input files for training and prediction purposes. The output shows one of four erosion classes, namely, shallow landslides, with four different probability thresholds.

### 4.1. Object-Based Image Analysis

Object-based image analysis (OBIA) combines a segmentation algorithm with classification techniques ranging from decision trees to various supervised machine learning algorithms which assign generated segments (or object primitives) to erosion classes. We used the software eCognition Developer (version 9.3.2) implementing a multi-resolution segmentation algorithm for grouping pixels with similar properties to object primitives. Input data consisted of aerial imagery (RGB), the excess green vegetation index, and information from the DTM and its derivatives (slope, aspect, and curvature). The object primitives contained information on their spatial, spectral, textural, and contextual properties based on all input data. Given these extracted feature sets, a random forest classifier was trained on manually selected samples in order to identify bare soil sites or sites with reduced vegetation cover. Subsequently, an additional decision tree was assigned specific erosion classes based on the typical appearances of objects previously identified containing bare soil or reduced vegetation cover. These erosion classes consist of shallow landslides, livestock trails, sheet erosion, and management effects. Note that the entire workflow needed to be performed on every input image to accommodate for varying image properties. Therefore, OBIA-labels for different input images can be considered to be obtained from independent models (i.e., differently calibrated settings). A detailed description of the workflow is presented in Zweifel et al. [7].

### 4.2. Neural Network Architecture

In this study, we make use of the U-Net architecture [35] illustrated in Figure 5. U-Net is a fully convolutional neural network which consists of a contracting part and an expansive part. See Section S1 in the Supplement for more details on the main components of the neural network which are used in the following description.

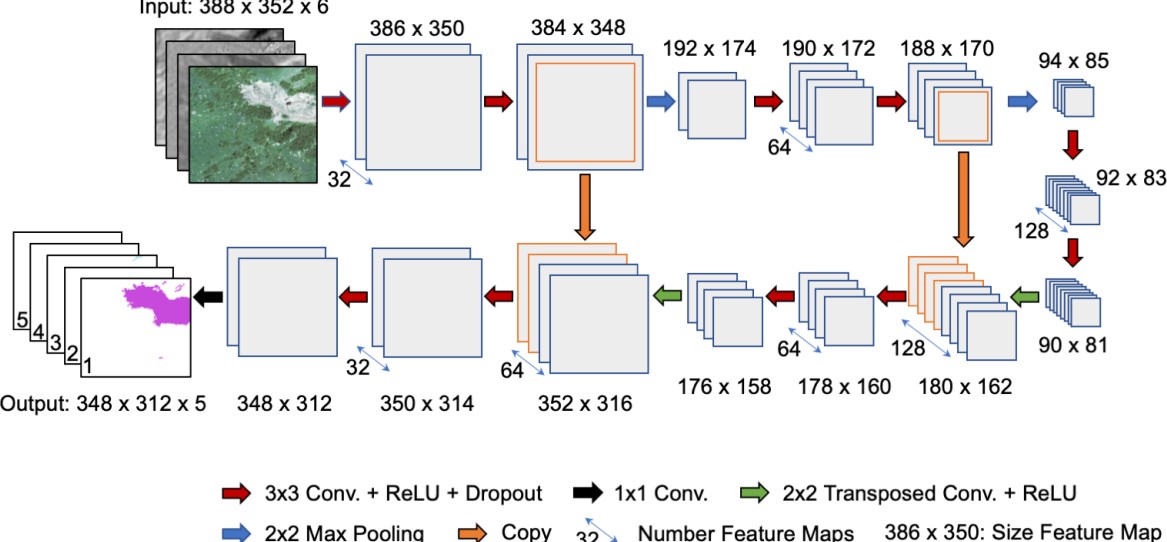

**Figure 5.** The employed U-Net architecture: In the first (upper) part, the input is contracted into a compressed representation (right). In the second (lower) part, the compressed representation is expanded into a segmentation map with pixel-wise class probabilities. The input consists of the inpuxt RGB image (three channels) and the DTM derivative maps for the aspect, curvature, and slope (one channel each). The resulting output provides a segmentation map for each considered class: Shallow landslides (indicated by 1 in the output), livestock trail (2), sheet erosion (3), management effects (4), and a class for non-assignable pixels (5).

In the contracting part (upper part in Figure 5), a sequence of two convolutional layers with ReLU activations followed by max pooling layer processes the input. With each max pooling application, the sizes of the resulting feature maps are halved, while the number of features is doubled for the subsequent convolutional layer. In the expansive part (bottom part), a sequence of transposed convolutional layers with ReLU activations followed by two convolutional layers and ReLU activations is applied to restore the original image size. Feature maps from the contracting part are appended to the feature maps obtained through the transposed convolutions to provide fine-detail features in the expansive part. Finally, a $1 \times 1$ convolutional layer followed by a pixel-wise softmax activation function provides the final segmentation output where each channel represents the segmentation map for the individual classes. The softmax function rescales the activations for each pixel to the $[0, 1]$ interval. More explicitly, for a pixel $f$ in the output map $F$, the softmax yields a prediction $p_c(f)$ which can be interpreted as the probability of pixel $f$ to belong to class $c \in \{1, \ldots, C\}$. The neural network is trained with the cross entropy loss which penalises incorrect class assignments with

$$-\frac{1}{N} \sum_{f \in F} \sum_{c \in C} y_c(f) \log(p_c(f)) \tag{1}$$

where $N = |F|$ is the number of pixels and $y_c(f)$ is the ground truth class assignment for pixel $f$, i.e., 1 if $c$ is the correct class and 0 otherwise. For any pixel $f$ in the input image, the softmax prediction $p(f) = (p_1(f), p_2(f), \ldots, p_C(f))$ provides the probabilities for the classes $c \in \{$shallow landslide, sheet erosion, livestock trail, management effect, non-assignable$\}$—e.g.,

$$p(f) = (\underbrace{0.55, 0.1, 0.2, 0.05}_{\text{erosion class probabilities}}, 0.1). \tag{2}$$

In addition to the four erosion classes, a class for non-assignable pixels is introduced which represents the class for all remaining (potentially ambiguous or vegetation covered and thus stable) objects.

U-Net provides pixel-wise class probabilities like in Equation (2) as the probabilistic output. In the following, for each erosion class we will refer to the full-probability result when only entries for the specific class of this output are considered without applying a threshold (e.g., the first entries for shallow landslides). For the final hard segmentation, we would like to obtain the dominant erosion class and apply different probability thresholds that control to which extent candidate segments are obtained. We only consider the erosion classes and identify the class with the largest probability for pixel $f$ as the dominant erosion class. If the selected erosion class probability does not meet the threshold, the respective pixel is considered as a background pixel. For example, in Equation (2), argmax $\{0.55, 0.1, 0.2, 0.05\}$ implies that shallow landslide is the dominant class and pixel $f$ is predicted to be a shallow landslide pixel with a probability of 55%. At a threshold of 0.5, the class probability exceeds the threshold and pixel $f$ is assigned to the shallow landslide class, while with a stricter threshold of 0.6 the pixel is considered to be a background pixel. With this kind of threshold segmentation, the final erosion class labels are obtained.

*4.3. Training Process*

In order to learn how to identify erosion sites, precise boundaries for the different erosion classes are required for training U-Net. Inadequate training labels can deteriorate the spatio-temporal generalisation capability of U-Net. In this study, we used high-quality training labels provided by the OBIA approach (see Section 3.3.1), and we considered the resulting erosion class areas as the ground truth segmentation in this investigation. To process the input images efficiently, we divided the aerial images into tiles of size $194 \times 176$ m which correspond to $388 \times 352$ pixels at 0.5 m resolution (2000, 2004) and $776 \times 704$ pixels at 0.25 m resolution (2010, 2013, 2016). The same is done for the maps of the DTM derivatives aspect, curvature, and slope.

Adjacent tiles overlap such that a 20 m (40 and 80 pixels, respectively) margin of one tile is contained in an adjacent tile. Figure 6 illustrates the resulting tiles for different years. The higher resolution tiles were down-sampled so that all input tiles are of size $388 \times 352$ pixels. No data augmentation was employed, as we expect object size and orientation (e.g., north/south exposure) to be relevant features. As described previously, U-Net was trained from scratch with tiles extracted from the training area of the years 2000 to 2013, with a total of 1292 training samples. A U-Net of depth 3 with initially 32 (root) filters was used (see Figure 5), resulting in 467,525 network parameters. The network was trained for 300 epochs with a batch size of 20, using the Adam optimizer [55] with a learning rate of 0.001 and a dropout rate of 0.1. We used TensorFlow version 1.10 [56] for our implementation which is based on the U-Net implementation by [57]. The full source code of our analysis pipeline is available under the GNU public license: https://github.com/bmda-unibas/ErosionSegmentation.

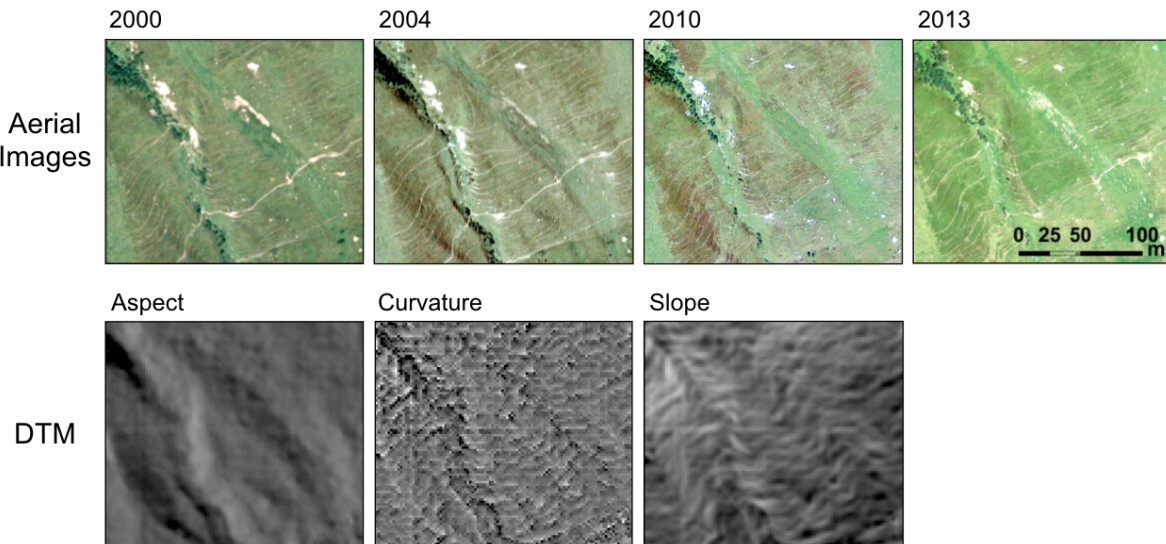

**Figure 6.** Example of input RGB images for training for the years 2000, 2004, 2010, and 2013 with a size of 194 × 176 m (corresponding to 388 × 352 pixels at 0.5 m resolution). The images show examples of eroded area on grassland slopes (livestock trails, shallow landslides). Below, the corresponding aspect, curvature, and slope maps are displayed (for all years the same DTM information is used). To obtain the samples, the aerial images of the respective years (Figure 2) and the DTM derivatives were divided into smaller tiles.

### 4.4. Details on the Evaluation

For the evaluation, only sites with an area of at least 4 m$^2$ were considered, which we treated as the minimum reasonable object size, and this is in line with the definition used in Zweifel et al. [7]. After choosing an appropriate probability threshold, the quality of the segmentation results was assessed with the precision score (producer's accuracy), recall score (user's accuracy), and their harmonic mean, the $F_1$ score. We considered objects which overlap in both the OBIA and U-Net results as true positives and weigh true positives, false positives, and false negatives by the areas of the respective segments. Ultimately, our goal was to evaluate the total degraded area on the held-out test area of the training years (2000, 2004, 2010, and 2013) and the validation year 2016 in comparison to the OBIA ground truth results. The emphasis here was to study the temporal trend and relative increase in degraded area as obtained from the different methods. We performed a linear regression to provide the linear trend over the time period from 2000 to 2016.

## 5. Results and Discussion

U-Net provides pixel-wise probabilities for each erosion class, which allows for assessing the certainty of predictions by studying the resulting heatmaps (see Figure 7 for an example). In practice, this rich information is further post-processed by applying a threshold on the pixel-wise probabilities to form well-delineated segments. In the following, we present both results on the (full-probability) heatmaps and results obtained with a selection of different probability thresholds. The latter enables a more direct comparison to the segmentation results obtained with OBIA. All results were obtained on the held-out test area (see Figure 2). Note that the data from 2016 was not used for training.

### 5.1. Segmentation of Soil Erosion Sites

The trained U-Net provides satisfying segmentation results which are demonstrated in Figure 7 for exemplary segments of shallow landslides and livestock trails. The heatmaps illustrate the full-probability output of U-Net and display the certainty in the class assignment (upper panel). By selecting different thresholds, hard class assignments can be achieved which lead to slightly different segment shapes depending on the threshold (lower panel). We selected thresholds of 0.2 and

0.8 to display the impacts of a wide range of probability thresholds on the delineation of segments. In general, choosing lower thresholds allows for the identification of a large number of potential erosion sites, while a higher threshold reduces the number of segments and also has an effect on the margins of these object, i.e., shrinks the segments to the most certain area. The probability threshold is a free parameter which can be chosen guided by application requirements or user preferences, or in our case to match baseline results (OBIA).

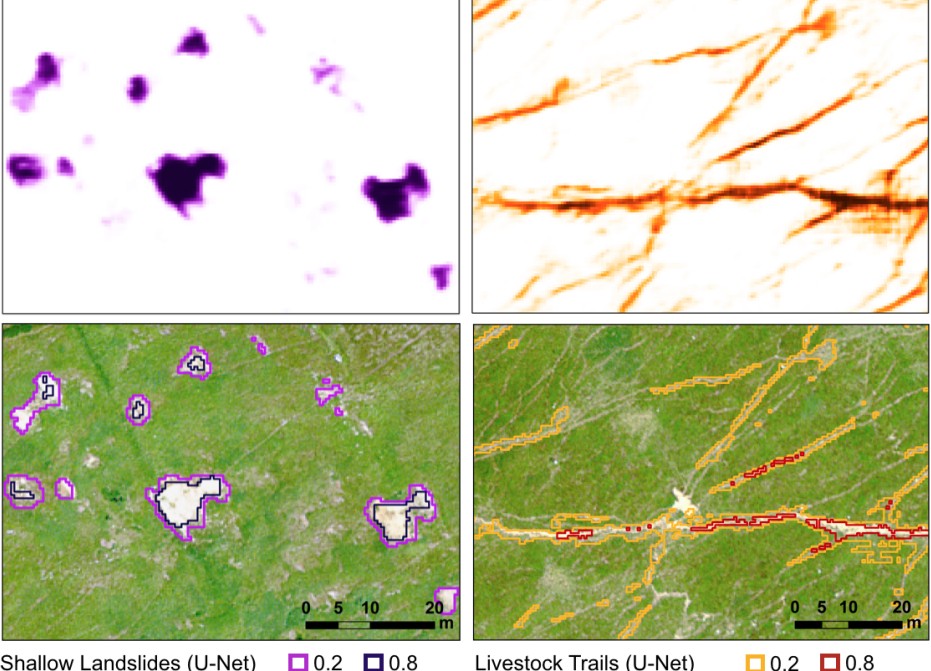

**Figure 7.** Visualisation of U-Net mapped shallow landslides (**left**) and livestock trails (**right**) for 2016. The lower panel shows segmentation results with different probability thresholds: the lighter colour indicates a lower probability threshold (0.2) and the darker colour indicates a higher probability threshold (0.8). Lower thresholds lead to larger and more numerous segments. For the same region (background omitted for better visualisation), the upper panel shows the full-probability heatmap output of U-Net: darker colours indicate higher probabilities.

In order to evaluate the accuracy of the proposed U-Net approach, we consider the OBIA results for 2016 as the ground truth baseline, which are independent of all other years, as OBIA was separately applied to the aerial image of 2016. For the comparison, we selected a threshold value of 0.3, as this led to the best agreement between U-Net and OBIA segments with respect to the total degraded area (see Section 5.3). OBIA relies on a dedicated, multi-resolution segmentation algorithm which provides clear objects to start with, which can then be classified. In contrast to OBIA, the U-Net approach does not have such a procedure and thus provides less control over segment shapes, as these are determined by pixel-wise thresholding. Consequently, there are cases in which both OBIA and U-Net identify areas as erosion sites but the boundaries of these objects might differ slightly. In that respect, our results for U-Net show that erosion sites with clear, unambiguous boundaries such as shallow landslides (and some very clear cases of livestock trails) generally have better overlaps with the OBIA baseline and contiguous objects are better identified (see Figure 8 on the left). Boundaries of more diffuse erosion sites predicted by U-Net show a slight mismatch with the OBIA baseline (see Figure 8 on the right). In these cases, the correct delineation of sites belonging to management effects or sheet erosion is in general a challenging task which is mirrored in the less accurate matching of the segmentation results from the different methods. Additionally, these erosion classes have similar appearances and are comprised of either bare soil or vegetation areas with strongly reduced vegetation cover which are prone to similar erosion processes (mainly erosion by water run-off), and they differ only in the origin

of the damage. Management affected sites are mostly located near the foot of the slope, are mainly used for the production of hay, and can show signs of heavy machinery usage. Sheet erosion, on the other hand, can be found throughout the entire valley and can be caused not only by livestock trampling and grazing, but also climate-related factors, such as drought, precipitation, and snow-melt [7,58–61]. Still, both methods are able to identify a great majority of overlapping objects. More quantitatively, we obtained scores for a threshold value of 0.3, as presented in Table 2.

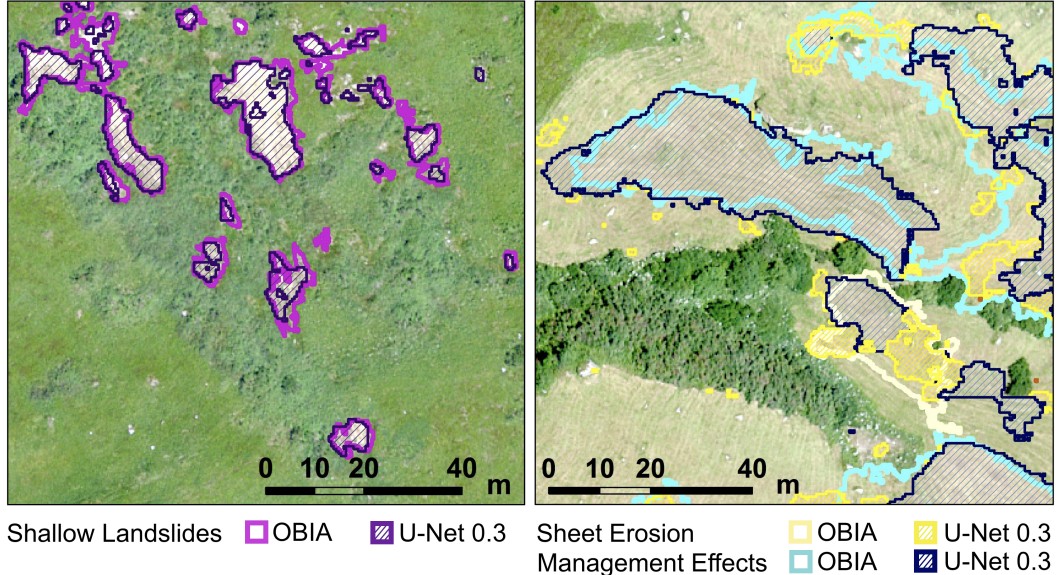

**Figure 8.** Comparison of segmentation results of OBIA and U-Net (probability threshold of 0.3) for the aerial image of the year 2016. This aerial image was not used during training of the U-Net model and depicted sections are located in the held-out test area. Lighter colours show OBIA results; darker colours (shaded) are results of U-Net.

**Table 2.** Scores for U-Net with a threshold value of 0.3 for the validation aerial image of 2016. U-Net results are compared to OBIA baseline results.

| Scores | U-Net |
|---|---|
| Recall | 84% |
| Precision | 73% |
| $F_1$ | 78% |

The precision score indicates that 73% of the predicted U-Net segments have corresponding OBIA segments, and about 27% of predicted U-Net segments do not directly correspond to any OBIA segments. On the other hand, the recall displays that 84% of the OBIA segments are maintained and the remaining 16% of OBIA segments are not identified by U-Net. Both these findings suggest that U-Net successfully identifies a majority of OBIA segments (recall score), but provides more segmented erosion sites than OBIA (false positives). Segments contributing to the 27% false positives still can be valid erosion sites which are not captured by OBIA, as it is known that OBIA tends to give a conservative estimate of the degraded soil [7]. Therefore, it is important to note that these scores mainly highlight the difference between U-Net segmentation with respect to the OBIA segmentation baseline, and it is possible that one method captures valid erosion sites which the other method misses (see example shown in Figure 9 on the right).

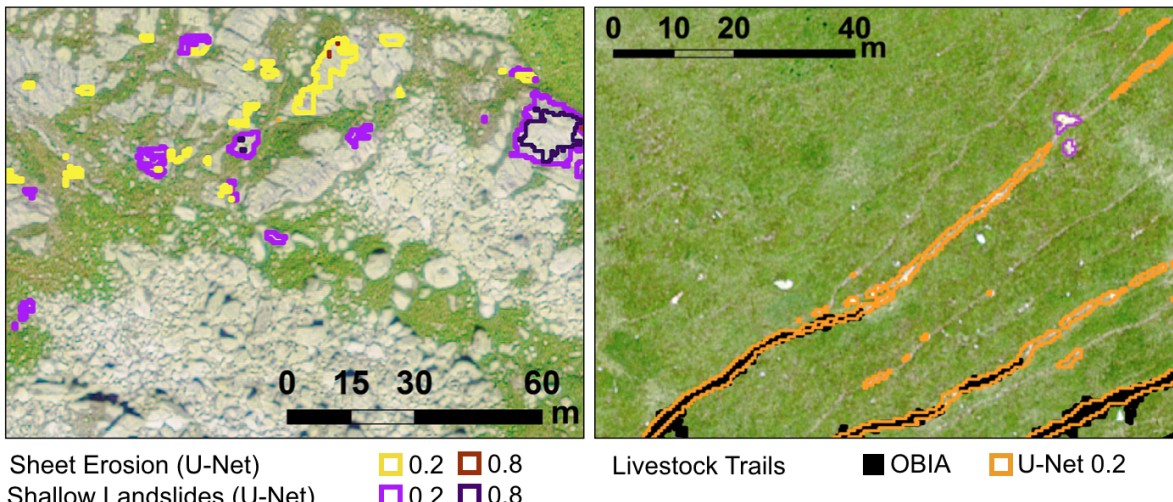

**Figure 9.** Examples of two different types of false positives: On the left-hand side, U-Net identifies some rock surfaces as sheet erosion (yellow) and shallow landslides (purple). For both erosion classes, thresholds of 0.2 and 0.8 are shown. Lower threshold choices are linked to more of such false positives. Depicted on the right-hand side are livestock trails with OBIA and U-Net (threshold of 0.2). Here, U-Net is capable of identifying more livestock trails correctly compared to OBIA.

Most cases of false positive predictions can be related to objects which are similar in appearance to the erosion classes, and the reason for misclassification can be recognised in many cases upon manual inspection. False positives are typically patches with rocks located at higher elevations which are classified as shallow landslides (see Figure 9 on the left), or varied classification of sites affected by management and sheet erosion. Nonetheless, singular rocks on grassland areas are successfully left unclassified. These kinds of disagreements are inherent to the U-Net approach, which attempts to identify regularities in the training data and thereby includes objects which share some similarities. In clear cases, such as very small object sizes or predictions at certain altitudes where a particular class of erosion phenomena is not expected, a post-processing step can address these erroneous classifications. Another way of avoiding segmentation ambiguities is to employ pre-processing steps to identify sub-regions of interest for target objects which share some kind of regularity in their appearance, for instance, in the shape of the objects [62]. For the purpose of this study, however, no pre-processing steps were used in the U-Net procedure to ensure objective comparison with OBIA.

*5.2. Threshold Selection*

In similar studies, the matter of threshold selection is usually not addressed or a fixed threshold value is used. This can be suitable for studies with binary output classes (e.g., [37,48]), but can also be problematic for gradual transitions of classified objects, as discussed by Kattenborn et al. [46]. Other studies employ deep learning approaches for classification of the object primitives in the OBIA framework where object boundaries are already well-defined [63–65]. In our setting, threshold selection can be used to adjust segmentation results in relation to pre-existing knowledge (i.e., segmentation results of other methods such as OBIA), which led to the best fit with a threshold selection of 0.3 for this study (with respect to the total degraded area; see Section 5.3). Additionally, varying thresholds may be applied to make necessary adjustments for different classes with varying appearances. As a standard comparison, a held-out data set of the ground truth segmentation required for training can be used to determine appropriate probability thresholds if necessary. In the absence of appropriate pre-existing knowledge or in cases where visual assessment is not possible, it is advisable to use a range of probability thresholds which capture a variety of segment estimations and assess uncertainty ranges of the estimates.

## 5.3. Trend Analysis of Soil Erosion Sites

In order to study the temporal trend in the extent of soil degradation, we applied U-Net to the series of five aerial images of the Urseren Valley between 2000 and 2016 (see Section 3.3.1). We compare the full-probability U-Net results and the results for the different thresholds to the baseline results of the OBIA approach in Figure 10. In the first case, the heatmap results are added up to form an estimate of degraded area per erosion class. The resulting outcomes of the full-probability U-Net output match the OBIA results closely with respect to the total degraded area. Due to their methodological differences, slight deviations in the segmentation results and the resulting (total) degraded area were expected. The same holds true for the U-Net results with a threshold of 0.3. This threshold was identified to exhibit the most suitable agreement with OBIA segmentation results with respect to the total degraded area. It can be observed that for validation year 2016, the OBIA and U-Net threshold 0.3 results agree very well (in the shaded area in right plot of Figure 10). As expected, the U-Net results display an increase in degraded area for decreasing thresholds. Nevertheless, in all considered U-Net results, the same temporal trends of decrease and increase from one year to another are observed, as in the OBIA baseline. This observation is also supported by the linear regression results, which in all cases provide similar linear temporal trends.

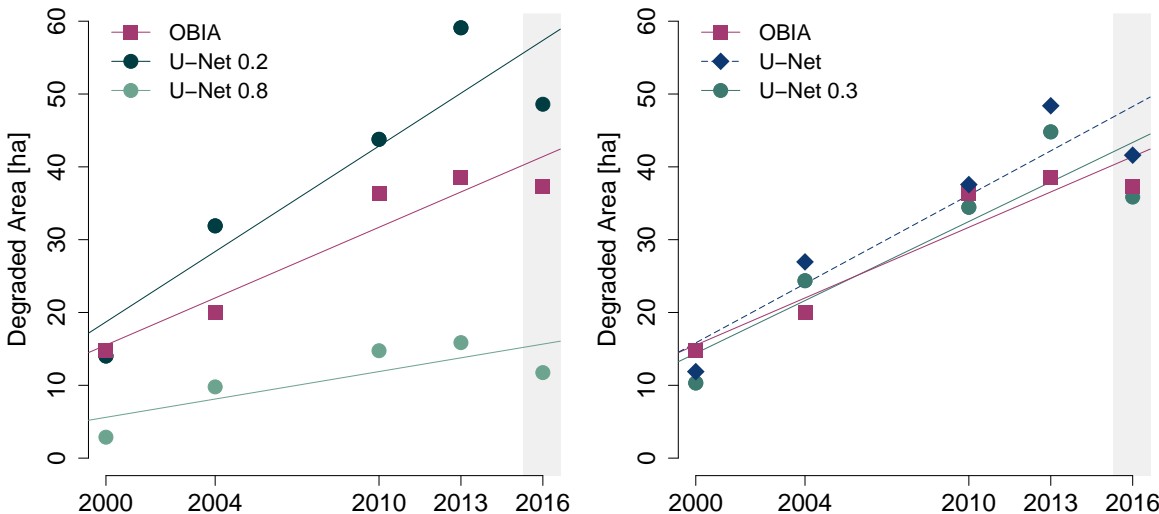

**Figure 10.** Linear trend of the total degraded area in the held-out test region (see Figure 2) as obtained with the OBIA and U-Net approaches. On the left, the results for a range of different threshold values are displayed; on the right the results for the suitable threshold value 0.3 and the full-probability results are given. Qualitatively, a similar increase or decrease of degraded soil in the individual years is retained in all models. The linear interpolation provides a similar temporal trend of increase in degraded soil in all cases. In particular, the full-probability and threshold 0.3 results of the U-Net approach show good agreement with the OBIA baseline. The linear trends with lower and higher thresholds surround the OBIA result. The years 2000 to 2013 provide a result on the spatial generalisation of U-Net (years used for training), while the result for 2016 (shaded column) in addition provides a temporal generalisation result (aerial image of 2016 was not used for training). Note that the OBIA approach needs to be trained on all aerial images.

In order to quantify the relative increase in degraded area, we consider the values for 2000 and 2016 obtained from the linear regression line. Again, the threshold dependency with respect to the total degraded area is observed (top panel in Figure 11). However, for the relative increase in degraded area (quotient of values for 2016 and 2000), the results become mostly independent of the selected threshold (bottom panel in Figure 11). To assess the statistical uncertainty of the linear regression fit and thus the relative increase, one standard deviation each of the fitted parameters

(slope and intercept) is considered to obtain the two most extreme linear trends which are possible within the uncertainty of the fitted parameters. This means the steepest and flattest linear trends with respect to one standard deviation in the parameters are identified, which leads to the error bars for the total degraded area as depicted in Figure 11. As the relative increase considers the ratio of these quantities, the error bars are relatively larger for the relative increase of degraded area. In particular, for a threshold of 0.8, the statistical uncertainty increases due to the comparably small degraded area detected. The obtained U-Net results show similar relative increases of degraded area which fall within the uncertainty range of each other depicting the statistical uncertainty in the linear regression fit (one standard deviation). The U-Net results are in good agreement compared to the baseline method, with an increase of 167% in the test region. This in turn is in line with the increase of $156 \pm 18\%$ reported in Zweifel et al. [7] for the full Urseren Valley, where $\pm 18\%$ depicts the estimated propagated error based on expert accuracy assessment (and not the statistical uncertainty in the linear regression fit). Importantly, it has been established that OBIA tends to underestimate the extent of degraded soil [7]. Therefore, the steeper relative increase obtained by the U-Net results is plausible and potentially reflects the increase of degraded area more accurately. Furthermore, the fact that the relative increases for the different probability thresholds coincide with each other within the statistical uncertainty of one standard deviation of the linear regression fit is further evidence for the applicability and robustness of the U-Net approach. Assessing the relative development of aggregated measures, such as the total area of degraded soil, is therefore less sensitive to the choice of threshold. The results on the linear trend (Figure 10) and the relative increase of total degraded area (Figure 11) highlight that the probabilistic output of U-Net aligns with the OBIA results very well, and to study these quantities by choosing a threshold, i.e., hard segmentation, is not required. In our investigation we assess predictions in the held-out test region (see Figure 2) for two validation cases: (i) testing the erosion site prediction of the test region for years for which conditions (colour, shading, vegetation, etc.) were available during training (2000–2013) and (ii) testing the predictions for a new year for which conditions were unknown during training (2016). In the first case, our results provide evidence that the trained U-Net transfers well to adjacent regions with similar conditions, as observed during training, and that shows the spatial generalisation capability of the U-Net approach. Furthermore, the latter validation case gives evidence of suitable erosion site segmentation with the U-Net approach in completely new aerial images with conditions not encountered during training, which in addition highlights the temporal generalisation capability of the approach.

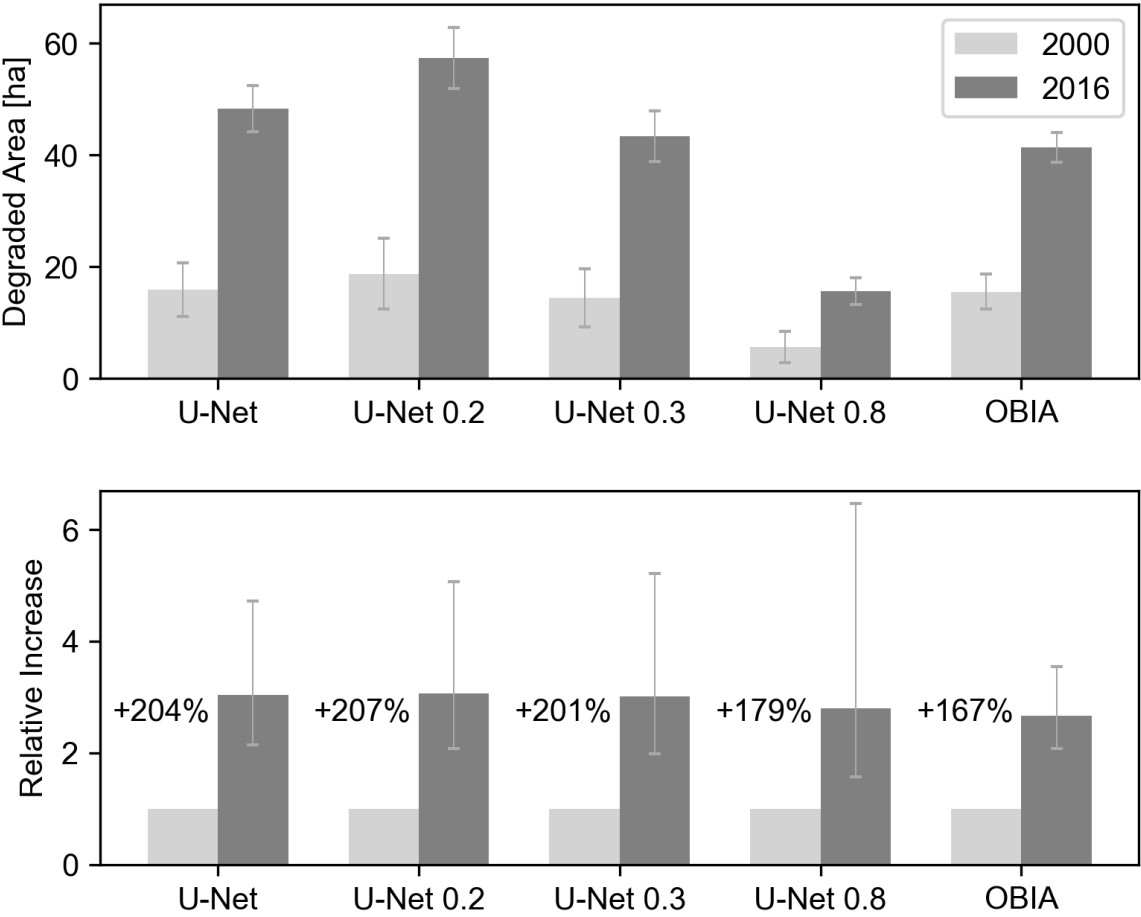

**Figure 11.** Comparison of total degraded area in years 2000 and 2016 for the baseline (OBIA) and the U-Net approach with different thresholds. The total degraded area was obtained from the interpolation results of each year (**top panel**). In all approaches, an increase of degraded area in the Urseren Valley is observed with threshold-specific differences in the total extent. However, the relative increase in degraded area (**bottom panel**) shows that assessing the trend of soil degradation can be done independently of the threshold, as all results fall within the statistical uncertainty of the linear regression fit. Note that the statistical uncertainty for U-Net 0.8 increases due to the comparably small total degraded area detected. The error bars depict the statistical uncertainty of one standard deviation.

For the individual erosion classes, we examine the results for the full U-Net model output and for a threshold of 0.3 (see Figure 12). Especially for sheet erosion and management effects, which contribute to a great amount of the total degraded area, the choice of 0.3 as a threshold for the hard segmentation is appropriate. In the case of livestock trails, the full-probability U-Net results capture the behaviour in the baseline more appropriately. The individual results highlight that an erosion-class-specific choice of the probability threshold can be reasonable in applications such as ours. We provide a result on such a mixture of thresholds for the linear trend for the years 2000 to 2016 in Supplementary Figure S2. The linear trend for the years 2000 to 2016 exhibits good agreement with the OBIA baseline (similar to Figure 10 on the right). Therefore, although the temporal development of aggregated measures is less dependent on the threshold, choosing different probability thresholds enables flexibility in the number of identified segments and segment boundaries in the U-Net approach. This is especially the case when examining the temporal development with regard to the degraded area per individual erosion class (Figure 12).

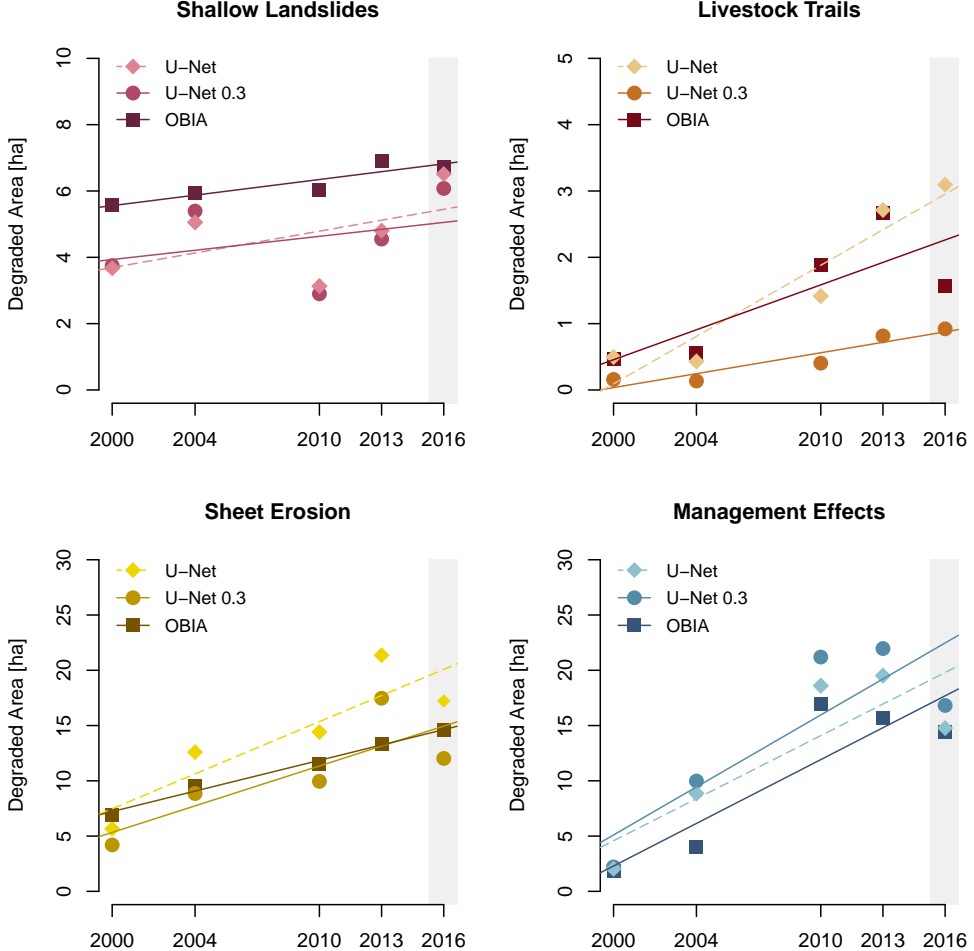

**Figure 12.** Mapped degraded area in the test region by erosion class for both the OBIA and U-Net methods (full-probability results and threshold value 0.3). Comparing the two methods, class-specific differences for the yearly degraded area and linear trends can be observed. Moreover, by selecting appropriate thresholds for each erosion class, similar linear trends in both methods can be attained (see Supplementary Figure S2). The years 2000 to 2013 provide a result on the spatial generalisation of U-Net (years used for training), while the result for 2016 (shaded column) in addition provides a temporal generalisation result (aerial image of 2016 was not used for training).

## 5.4. Deep Learning and OBIA

Deep learning methods for similar applications are predominantly trained with manual labels, and often the objects of interest are precisely defined, such as roads, buildings, or damaged trees in forests [37,43,47]. In our application, the objects are less clearly defined, and some of the segment boundaries concerning both the mapped and omitted areas might be more disputable. The boundaries of objects are often ambiguous due to smooth transitions, especially for erosion sites with reduced vegetation cover. Imprecise delineation of the objects of interest negatively impacts the generalisation capability and applicability of deep learning techniques, and can potentially be a limiting factor for this kind of approach. In particular, it can have a detrimental effect on the accuracy of the U-Net approach if the ground truth misses a great number of relevant objects. Therefore, we do not rely on manual labels of the objects of interest, which might suffer from subjective assessments, require labour-intensive work, and usually are unable to achieve pixel-level precision. Instead, we showcase that any kind of segmentation technique, such as OBIA in our study, can be used as a basis to provide training data to

successfully employ a convolutional neural network for segmentation of natural features, such as the erosion sites in our application.

Similar studies have compared OBIA to deep learning approaches for the detection of landslides on remotely sensed data with the goal of enabling large-scale analysis. In Prakash et al. [66] the comparison was done on the basis of landslide inventories. A study of different machine learning and deep learning methods was conducted by Ghorbanzadeh et al. [67], who used field observations with manual corrections as the ground truth segments. These studies show that deep learning approaches improve segment detection by comparison of the segmentation performances of the different methods. In our study, we leverage the fact that OBIA is a well-suited approach for segmentation tasks on small scales, and thus derive our baseline trends and the ground truth segments from it. Other work like the detection of shrubs on high resolution satellite imagery by Guirado et al. [62] similarly shows that CNN approaches can outperform OBIA in certain cases. That study relied on manually delineated ground truth segments and used dedicated pre-processing steps to identify regions of interest to perform classification of candidate patches. Combining OBIA and CNN approaches was also studied with regard to using CNNs in the classification step of the OBIA framework [63] or using features learned by the CNNs to improve inputs to the OBIA workflow [68]. In our study, OBIA provides the necessary high-quality ground truth segmentation, but our workflow is not bound to OBIA, and any other reliable approach can be used for this too.

The presented results of this study substantiate that the U-Net approach can perform on a par with OBIA. Moreover, the transferability to new data, the insensitivity of trends in aggregated measures to threshold selection, and the flexibility of fitting the U-Net results to existing knowledge or competing segmentation methods—apart from manual inspection of segmentation results—render the proposed approach advantageous for a great variety of applications. Furthermore, large-scale analysis is facilitated by improved running times. For training and prediction, an Nvidia GeForce Titan X Pascal GPU was used. In our study, training required a running time of approximately 6.5 h, while the prediction for the full Urseren Valley took 12 min. This is a significant improvement over the semi-automatic OBIA approach, which takes up to a few days to achieve satisfying results for the Urseren Valley. For large-scale studies (e.g., alpine-wide analysis) the process can efficiently be parallelised using several GPUs, resulting in even faster prediction times.

## 6. Conclusions

While OBIA is the state-of-the-art approach for mapping objects on remotely sensed images, it suffers from limitations that render this approach unsuitable for larger-scale studies. High-quality segmentation results come at the expense of a lack of transferability of parameter settings from one input image to another, manual adjustments, and a need for expert knowledge in applying the method to the specific task which together lead to long processing times. In particular, the first aspect generally hinders OBIA in a predictive setting for new images. To overcome these shortcomings and enable large-scale analysis, we compared OBIA to a fully convolutional neural network approach which learns relevant features for segmentation by itself and thereby emulates some of the expert knowledge necessary to apply OBIA. We demonstrated that the U-Net approach is capable of performing as well as OBIA with respect to identifying trends in the spatial and temporal development of degraded soil, and can therefore replace OBIA in large-scale studies. Spatial patterns and temporal trends of both methods agree well; nevertheless, some generated segmentation results might partially not overlap ($F_1 = 78\%$). Specifically, we show that U-Net (threshold 0.3) provides a potentially more accurate relative increase of total degraded area in the Urseren Valley than the more conservative estimates of OBIA (201% vs. 167%). This novel approach allows for individual threshold choices for the most successful representation of ongoing soil erosion processes. This is typically possible if some prior knowledge about erosion processes and the spatial extent of degraded soil is available, or if visual assessment is feasible, to which probability thresholds can be calibrated. In our study, we made use of training labels generated with OBIA. However, any kind of (high-)quality training labels can be

used, and the U-Net erosion site segmentation is not limited to combined use with OBIA. In summary, we show that with our approach we can perform erosion site prediction close to similar approaches such as OBIA which provide accurate segmentation results on small scales. A particular strength of the proposed approach is that similar trends are achieved with a more efficient, automatic, and objective method for mapping erosion sites. We require the U-Net approach to be trained only once and obtain much better transferability of the method to new images. Moreover, the approach is insensitive to the threshold choice with respect to trends of aggregated measures, and the improved running times make large-scale analysis of soil erosion is Swiss alpine grasslands feasible.

Still, our model is only as good as the training data; i.e., high-quality training data are important for adequate U-Net performance. Future studies should include a variety of different sample regions to incorporate relevant erosion-type-specific conditions during training (e.g., orientation of erosion sites). Furthermore, U-Net can use as many layers of information as required. A unique feature of fully convolutional neural networks is that inputs of any size and any number of channels can be used, i.e., RGB images with DTM derivatives. Additional maps can be easily incorporated (see Figure 5), which might include more information, such as environmental properties or images with additional spectral information. In that regard, U-Net has the advantage of continual learning; i.e., it can be trained further to incorporate conditions of completely new regions and erosion-type-specific properties. Generally, the U-Net model can be employed in a similar fashion for other segmentation tasks in remote sensing and other inputs, such as UAV or satellite imagery. The requirement for the input data is that the spatial resolution allows for identifying the target objects well enough.

**Supplementary Materials:** The following are available online at http://www.mdpi.com/2072-4292/12/24/4149/s1 . Section S1: Details on the neural network architecture, Figure S1: Illustration of the max pooling and convolution operation, Figure S2: Mixed thresholds for trend analysis.

**Author Contributions:** Conceptualisation, M.S., L.Z., V.R., and C.A.; methodology, M.S. and L.Z.; software, M.S.; validation, M.S. and L.Z.; formal analysis, M.S. and L.Z.; investigation, M.S. and L.Z.; data curation, M.S. and L.Z.; writing—original draft preparation, M.S. and L.Z.; writing—review and editing, M.S., L.Z., V.R., and C.A.; visualization, M.S. and L.Z; supervision, V.R. and C.A.; project administration, M.S. and L.Z.; funding acquisition, V.R. and C.A. All authors have read and agreed to the published version of the manuscript.

**Funding:** This research was funded by Swiss National Science Foundation with the grant number 407540 167333 as part of the Swiss National Research Programme NRP 75 "Big Data".

**Acknowledgments:** The authors would like to thank the Swiss National Science Foundation for supporting the research. Calculations were performed at sciCORE (http://scicore.unibas.ch/) scientific computing core facility at University of Basel. We also want to acknowledge Swisstopo and MeteoSwiss for providing the data sets we used. Furthermore, we would like to thank the anonymous reviewers for their comments and suggestions which helped us to improve the paper.

**Conflicts of Interest:** The authors declare no conflict of interest.

## Abbreviations

The following abbreviations are used in this manuscript:

| | |
|---|---|
| OBIA | Object-based image analysis |
| DTM | Digital terrain model |
| RGB | Red, Green and Blue spectral bands |
| CNN | Convolutional Neural Networks |
| U-Net | Name of Convolutional Neural Network architecture |
| GPU | Graphics Processing Unit |
| UAV | Unmanned Aerial Vehicle |

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
