# Peer review of "Identifying Soil Erosion Processes in Alpine Grasslands on Aerial Imagery with a U-Net Convolutional Neural Network"

_remotesensing, doi:10.3390/rs12244149_

Round 1

Reviewer 1 Report

The topic of the paper is very interesting and relevant for this Remote Sensing special issue. I fully appreciate the excellent work conducted by the authors for developing the proposed U-Net Convolutional Neural Network model as well as for the meticulously preparation of the manuscript.    

Nonetheless, there are some minor aspects that I would have liked to be addressed, namely:

  • Since the access to aerial imagery is usually not open and free, would it be possible to tailor the proposed U-Net model for satellite imagery? Currently, as it is commonly known, there are some important sources of open and free satellite data. The Copernicus Programme itself provides a wealth of satellite imagery that is widely used by scientific and commercial users.

  • In today's Earth Observation developments, the volumes of data are colossal. Could the proposed U-Net model be adapted and used for extremely large datasets?

  • Considering that the proposed U-Net model was tested on five aerial images, could you please comment on its expected performance in case of larger remote sensing datasets?

  • Could the proposed U-Net model be adjusted for other types of applications besides erosion mapping?

As a suggestion only, the answer to (some of) these questions could be included in the Future work, under Conclusions.

Author Response

Thank you for your appreciation and the great feedback.

  • "Since the access to aerial imagery is usually not open and free [...]"

    This is a very important point you make. In general, the U-Net is not limited to a particular kind of input data. Typically, the size of the expected segments / objects determines what kind of input source and pixel resolution is required. In our case it is important that the spatial resolution is high enough, as certain objects are very small / narrow (e.g. livestock trails). An alternative would be the use of UAVs, which are capable of providing high resolution images. Of course, if the features you want to map are large enough also satellite images can be used (e.g. open access Sentinel-2 data with 10 m resolution). In future, when the spatial resolution of open access satellite imagery increases, this will be a great source of data for soil erosion mapping, but we are not quite there yet. We have addressed this point in the manuscript in lines 467-469.

  • "In today's Earth Observation developments, the volumes of data are colossal [...]" & "Considering that the proposed U-Net model
    was tested on five aerial images [...]"

    We would like to address the questions concerning large datasets jointly. In our manuscript we show that the U-Net approach trained on a comparably small training set performs very well and generalises / transfers to adjacent regions and new aerial images. Having larger (and more diverse) datasets at one's disposal for training will benefit the generalisation capability even more. Providing adequate training data would potentially be the most time consuming part but could be performed in the fashion outlined in the manuscript. What concerns the prediction, i.e. providing erosion segments for a new input, our trained U-Net required about 12 minutes to provide the segmentation results (threshold 0.5) for the case study area of about 26 km2 on a recent GPU. Extrapolating these findings, the segmentation of the Swiss Alpine region (say 60% of Switzerland, i.e. 0.6*40000 km2 = 24000 km2) would amount to about 8 days on a single GPU (24000 km2 * 12 min/26 km2 = 11077 min = 7.7 days). As the prediction process can be parallelised by using several GPUs, this running time can be further reduced. We want to emphasise that this is a model calculation based on our findings which illustrates that the proposed approach indeed can scale to large datasets. Our future work focuses exactly on these kind of large-scale applications and quantification.

  • "Could the proposed U-Net model be adjusted for other types of applications besides erosion mapping?"

    There are many applications in remote sensing using the U-Net model and we have listed some examples in the introduction (lines 67-70). So, in general, the U-Net approach can be used for an kind of segmentation task. However, in most cases there are preexisting datasets that can be used for training. For applications where there is no training data this might pose a problem: the application of the U-Net is therefore limited to availability of training data or the willingness/possibility to prepare own suitable training data. The more precise the training data is, the better the results will be with the U-Net model. For this reason we used data generated with Object-based Image analysis (from our previous study, reference [7] in the manuscript), which provides very good object segmentation. We trained the U-Net model to identify soil erosion based on the four erosion classes mentioned in the paper. But the U-Net can be used for any application (given that objects are adequately visible on images) when trained with the appropriate dataset (lines 468-469).

  • "As a suggestion only, the answer to (some of) these questions could be included in the Future work, under Conclusions."

    Thank you for your input! We have adapted the manuscript to address some of your comments (lines 467-469).

Reviewer 2 Report

In this manuscript, U-Net Convolutional Neural Network was applied to detect erosion types using training data generated from the object-based image analysis. Multi-temporal aerial imagery and digital terrain model derivatives were used as input datasets to classify erosion types: shallow landslides, livestock trails, sheet erosion, and management effects. Optimal U-Net threshold was selected based on the best agreement between U-Net and OBIA segments. I have following comments for your consideration. 

What would be the ranking of input datasets in terms of significant contribution to the U-Net result? 

Magnitude and duration of landscape changes (land disturbance agents and vegetation regeneration types) that are different from trained model would influence the result. Why land change derivatives were not used in the training and prediction processes?

Would Dimensionality reduction (e.g., PCA) of aerial imagery improve accuracy of U-Net result?

Author Response

Thank you for your insightful comments and suggestions!

  • "What would be the ranking of input datasets in terms of significant contribution to the U-Net result?"

    This is an interesting point which is raised. Unfortunately, and to the best of our knowledge, there is a lack of methods which would allow quantifying or ranking the importance of input images or input features for neural network approaches such as the U-Net. However, initially we conducted experiments where the U-Net was trained only on the RGB input images, i.e. without the DTM derivatives slope, curvature and aspect. Although this provides already acceptable segmentation results, our general finding was that incorporating information on the DTM derivatives as an additional input improved the segmentation results. This finding is also in line with the results of our previous study (reference [7] in the manuscript), where the feature importance of the Random Forest classification in the OBIA workflow was considered. However, within our project we are developing approaches how to visualise the relevant regions in the U-Net input which eventually might allow ranking the inputs to address questions such as posed by the reviewer.

  • "Magnitude and duration of landscape changes [...]"

    If such data is available at appropriate spatial and temporal resolution, this could potentially be a useful input. In the manuscript, we allude to that any kind of spatial information can be included in this U-Net approach, which could potentially improve the accuracy of the model (lines 461-465). However, in this study, we wanted to focus on data which is typically available on large scales (RGB images and DTM). This allows the method to be transferable to other regions, without depending on datasets which may not be available everywhere.

  • "Would Dimensionality reduction (e.g., PCA) of aerial imagery improve
    accuracy of U-Net result?
    "

    This is an interesting idea which is raised. One can view the compression part
    of the U-Net (see figure 5 in the manuscript) as performing dimensionality
    reduction (at least spatially). However, approaches do exist which combine the
    idea of the Variational Autoencoder (which can be considered to perform PCA
    like dimensionality reduction) with the U-Net and, in some applications, show
    improved and more accurate segmentation results. Indeed, this is a direction
    which we currently investigate also in the context of soil erosion segmentation.

Reviewer 3 Report

Dear Authors,

  the paper is very clear, well designed, and easy to read. I would just suggest to add the following citation to your introduction related to the U-Net uses: https://www.mdpi.com/2072-4292/12/22/3685/htm

A part from this minor revision, I recommend the publication.

Author Response

"Dear Authors, the paper is very clear, well designed, and easy to read. I would just suggest to add the following citation to your introduction related to the U-Net uses: https://www.mdpi.com/2072-4292/12/22/3685/htm. A part from this minor revision, I recommend the publication."

Thank you for your comments! We have included the suggested paper in the introduction (line 70).